# Supported Imidazolium-Based Ionic Liquids on a Polysulfone Matrix for Enhanced CO_2_ Capture

**DOI:** 10.3390/polym14224865

**Published:** 2022-11-11

**Authors:** David Domingo Huguet, Aitor Gual, Ricard Garcia-Valls, Adrianna Nogalska

**Affiliations:** 1Eurecat, Centre Tecnològic de Catalunya, Unitat de Tecnologia Química, C/Marcel·lí Domingo, 2, 43007 Tarragona, Spain; 2Faculty of Chemistry, Universitat Rovira I Virgili, C/Marcel·lí Domingo, 1, 43007 Tarragona, Spain; 3Department of Chemical Engineering, Universitat Rovira I Virgili, Av. Països Catalans, 26, 43007 Tarragona, Spain

**Keywords:** CO_2_ capture, ionic liquids, supported ionic liquid, gas adsorption

## Abstract

The present work demonstrates the potential for improved CO_2_ capture capabilities of ionic liquids (ILs) by supporting them on a polysulfone polymeric matrix. CO_2_ is one of the main gases responsible for the greenhouse effect and is a focus of The European Commission, which committed to diminishing its emission to 55% by 2023. Various ILs based on combinations of 1-butyl-3-methyl- imidazolium cations and different anions (BMI·X) were synthesized and supported on a polysulfone porous membrane. The influence of the membrane structure and the nature of ILs on the CO_2_ capture abilities were investigated. It was found that the membrane’s internal morphology and its surface characteristics influence its ILs sorption capacity and CO_2_ solubility. In most of the studied configurations, supporting ILs on porous structures increased their contact surface and gas adsorption compared to the bulk ILs. The phenomenon was strongly pronounced in the case of ILs of high viscosity, where supporting them on porous structures resulted in a CO_2_ solubility value increase of 10×. Finally, the highest CO_2_ solubility value (0.24 mol_CO2_/mol_IL_) was obtained with membranes bearing supported ILs containing dicarboxylate anion (BMI.MAL).

## 1. Introduction

The continued increase in the world population and its dependence on fossil fuels as an energy source has increased the atmospheric carbon dioxide (CO_2_) concentration. This steady rise in anthropogenic CO_2_ emissions from the beginning of the industrial era has increased atmospheric CO_2_ from ~280 ppm in 1750 to ~415 ppm in 2020 [1]. CO_2_, among others, is considered a greenhouse gas (GHG) because of its capacity to absorb the infrared range of electromagnetic radiation. This radiation comes from Earth’s surface after the irradiation of sunlight, avoiding the dissipation of this energy through the atmosphere to space, leading to an increase in the global temperature. The Intergovernmental Panel on Climate Change (IPCC) set the limit on the rise in global temperature to no more than 1.5 °C by the end of the century. To achieve this purpose, a drastic reduction in CO_2_ emissions is required until the total decarbonization of the industry by 2050. Taking a look at the principal industrial CO_2_ emitters, refineries, power plants operated by fossil fuels, steel-producing plants, cement plants, and the chemical industry are the main entities responsible for the total greenhouse gas emissions [2,3].

Significant research has been carried regarding the mitigation of CO_2_ emissions using various ideas, such as its capture or sequestration in solid absorbents using porous adsorbents [4].

Storage in underground reservoirs has been proposed as a principal solution, but this can only alleviate the problem without being a final solution. Moreover, issues of underground leakage can be possible, traduced to soil contamination and acidification due to the acidic nature of the gas, becoming an even bigger problem [2]. For this purpose, different strategies have been studied in recent years; for example, the use of liquid absorbent solutions has been explored. The liquid sorbent must exhibit certain characteristics: (1) enhanced CO_2_ sorption due to the combination of non-polar domains with high CO_2_ solubility (CO_2_ adsorption) and polar domains with reactivity towards the acidic molecule of CO_2_ (CO_2_ absorption), (2) moderate viscosity to facilitate CO_2_ diffusion, (3) chemical compatibility with the system components, and (4) low vapor pressure to avoid the loss of the solvent by evaporation [5]. The solvents used at industrial levels are based on basic aqueous solutions, the classical choices being alkanolamines, such as ethanolamine, and potassium hydroxide solutions. These aqueous solutions possess high alkalinity, reacting very quickly with the acidic molecule of CO_2_ and forming carbamates or carbonates, depending on the nature of the reactant [6,7]. However, these absorbents present different problems, such as their corrosive nature, the aqueous base of the solution making them volatile, and the high amount of energy required for the post-capture CO_2_ desorption from the system in order to use the CO_2_. This desorption is not as chemically favored compared to the sorption process, so a high amount of energy is required to desorb the CO_2_ and regenerate the sorbent solution for reuse [6]. To overcome these problems, new solvents, such as ionic liquids (ILs) or deep eutectic solvents (DES), are studied for this purpose because of their high solubility of gases. Among them, the use of ILs has been extensively reported. By definition, ILs are salts with a melting point below 100 °C representing a new class of solvents with a nonmolecular ionic character [8]. Different types of ILs have been reported, including completely inorganic ILs or ILs made up of a combination of organic cation and organic or inorganic anions [9,10,11]. Among their excellent properties are their negligible vapor pressure, high thermal stability, non-flammability, and stability against air and water, as well as their remarkable tuneability and versatility by simply varying the combination of cations and anions [12,13].

However, the use of large amounts of ILs is undesirable due to parameters related to the process engineering (i.e., viscosity), economic factors (high cost), and sustainability. To overcome these issues, the development of supported ionic liquid systems (SILS) has been extensively reported in the literature, with great success. In many cases, the materials obtained displayed the advantageous properties of the pure ionic liquid systems by using much smaller IL amounts supported onto a second material [14].

The support must be inert to ILs, so as not to alter its CO_2_ selectivity, and be mechanically stable. The most suitable candidates, in this case, are polymeric membranes. Polymers are very interesting materials for use as the matrix for the solid support of CO_2_ absorbers due to the availability of a variety of monomers, polymerization techniques, and possible additives to ensure the desired chemical and mechanical properties. One of the most used technics for porous polymeric membrane (PMs) preparation is the phase inversion process. For this procedure, a viscous solution of the desired polymer and a suitable solvent is prepared and cast onto a surface. Finally, it is immersed in a non-solvent bath where the polymer precipitates in the form of a porous flat sheet [15]. One of the characteristics of this procedure is that the internal morphology and the porosity of the obtained membrane can be controlled by preparation conditions (concentration, solvents, non-solvents, temperature, humidity) [16], making it possible to obtain the desired internal structure of the membrane for its application. The only requisite of the procedure is that the solvent and non-solvent must be miscible liquids.

The polymeric materials commercially available and mainly used for the membrane preparation can be seen in Figure 1. Among them, polysulfone (PSU) has been chosen for this work due to its excellent properties, including its: (1) mechanical strength; (2) thermo-stability; (3) chemical stability; (4) relative hydrophobicity; (5) ease of handling [17,18,19].

Previous studies have reported good sorption result using the approach of preparing SILs in a polymeric PSU matrix as CO_2_ sorption materials [20,21,22]. In this study, we developed the synthesis of different ILs based on combinations of 1-butyl-3-methyl- imidazolium cations and different anions (BMI·X) for use in the preparation of novel SILs on a polymer porous matrix. Thanks to this approach, we can increase the CO_2_ capture rate due to an increased gas/liquid contact area compared to that of bulk ILs and, at the same time, decrease the amount of ILs to be used in this application. Moreover, a solid adsorbent is easier to handle. These IL structures have been selected according to their reported values of CO_2_ solubility (Figure 2) (i.e., experimentally measured CO_2_ molar fraction in an IL solution of 0.20 to 0.60) [11].

## 2. Materials and Methods

### 2.1. Materials

Polysulfone (PSU, Mw 35,000 Da in transparent pellet form, Sigma Aldrich, St. Louis, MO, USA), *N*,*N*-Dimethylformamide (DMF, 99% Fisher, Loughborough, UK), and 1-Methyl-2-pyrrolidone (NMP, 99% Acros Organics, New Jersey, NY, USA) were used for membrane preparation.

For the synthesis of 1-butyl-3-methylimidazolium chloride, 1-chlorobutane (99% Acros Organics, Geel, Belgium), 1-methylimidazole (99% Sigma-Aldrich, St. Louis, MO, USA), acetonitrile (CAN, 99.8% Fisher, Loughborough, UK), and ethyl acetate (99.9% VWR, Fontenay-Sous-Bois, France) were used. The task-specific ILs were prepared by the ion exchange process using Amberlite IRA-402(Cl) ion exchange resin (Alfa-Aesar, Kandel, Germany), Sodium hydroxide (NaOH, 97% Fisher, Geel, Belgium), and acidic compounds: benzoic acid (99% Acros Organics, New Jersey, USA), pivalic acid (99% Sigma-Aldrich, Steinheim, Germany), DL-proline (99% Sigma-Aldrich, St. Louis, USA), formic acid (99% Merck, Darmstadt, Germany), and malonic acid (99% Alfa-Aesar, Kandel, Germany) 

The membrane gas solubility was evaluated with CO_2_ (99.995%) and N_2_ (99.999%) gases purchased from Linde (Barcelona, Spain).

### 2.2. Ionic Liquids Preparation and Characterization

Ionic liquids have been prepared using procedures adapted from the literature [23]. In a round bottom flask with a double neck, 1-chlorobutane (2.85 mol, 267 g) and 1-methylimidazole (2.85 mol, 237 g) were mixed in a molar ratio of 1:1. The solution was heated under reflux at 90 °C for 72 h while stirring. Next, the unreacted reagents were evaporated under vacuum at 50 °C, and the obtained solid was purified by recrystallization using acetonitrile (500 mL) as a solvent and ethyl acetate (1.5 L) as a crystallization phase. A white crystalline solid was obtained as a product, which was then filtered and dried under vacuum, leading to 1-butyl-3-methylimidazolium chloride (BMI.Cl) with a 90% yield (2.54 mol, 445 g).

Next, the selected ionic liquids were synthesized by an anion exchange using adapted protocols [11]. Using a glass column (3 cm × 50 cm) with 75 g (17 cm) of a Amberlite IRA-402(Cl) ion exchange resin, the chlorine anion of the BMI.Cl was exchanged using conjugated bases of acidic compounds, listed in Table 1, with 3 different groups of compounds; (1) carboxylic acids, (2) dicarboxylic acid, and (3) carboxylic acid with amine compounds. The general procedure for the IL preparation is described below. First, to ensure that all the active sites of the resin were free of chlorine anions, a 1 M solution of NaOH in 200 mL of milliQ^®^ water was passed through the column. After that, 1 L of milliQ^®^ water was passed to eliminate the excess of hydroxy anions. The procedure was monitored by the pH measurement of the water passed through the resin, resulting in a neutral pH at the end of the process. Then, 0.1 M of 1-butyl-3-methylimidazolium chloride in 500 mL of milliQ^®^ water was passed through the column, with an additional 250 mL of milliQ^®^ water, to wash out the remaining ionic liquid. Later, the calculated weight of the desired acidic compound was added directly to the aqueous solution and stirred in a 1 L Erlenmeyer for 48 h to ensure the pH equilibrium and the neutralization of the (BMI·OH) by the acid and the formation of the corresponding (BMI·X). Finally, (BMI·X) was dried under reduced pressure, using a rotavapor for 5 h at 80 °C, and using the Schlenck line system for 12 h at 50 °C.

Nuclear Magnetic Resonance (NMR) was used to determine the composition of the synthesized ionic liquids. Monodimensional (^1^H) experiments were performed in a Bruker Avance Neo 400 MHz spectrometer using deuterium oxide (D_2_O) as a solvent. Chemical shifts (ppm) were provided relative to trimethyl silane (TMS) in ^1^H NMR (16 scans). All ionic liquids purities (>99%) were determined using ^1^H NMR after drying and before its further use.

All ILs were first dried under high vacuum for 4 h at 60 °C before determining the density. Then, using an electronic pipette (Sartorius Picus NxT Electronic Pipette, 50–1000 μL, Goettingen, Germany), 1 mL of IL was collected, and its weight was measured using an analytical scale in triplicate to obtain the ionic liquid density (g/mL). The same procedure was followed for 75% *w*/*w* in milliQ^®^ solutions of ILs, where water was added to dry ILs.

Viscosity measurements were carried out using an IKA Rotavisc Lo-Vi Complete (Staufen, Germany), with an extender connector Vols 1.11, a Spindle Vol SP-6.7, and a sample chamber Vol-C-RTD-1. The viscosity of the ILs solutions used for membrane modification (75% *w*/*w* ionic liquid in milliQ^®^ water) was measured in triplicate. All measurements were carried out at room temperature the same day to ensure similar temperatures for all solutions (21.5–22 °C).

Surface tension was determined using a pendant drop system with a Dataphysics OCA 15EC (Filderstadt, Germany). Ionic liquid 75% *w*/*w* solutions in milliQ^®^ water were placed in a Hamilton de 500/gt syringe (Filderstadt, Germany), and the maximum droop volume was achieved using a manually controlled continuous flow dispenser. All determinations were carried out in triplicate at room temperature using a digital image created by SCA software (Filderstadt, Germany) included in the apparatus. As a reference, the syringe diameter was obtained, and the density of the solution was determined before measurements.

### 2.3. Membrane Preparation, Ionic Liquids Supporting Methods, and Characterization

A 20% wt. solution of polysulfone in DMF (D_Membranes) or NMP (N_Membranes) was prepared by stirring for 48 h. The obtained solution was kept for 24 h without stirring for its degasification and the removal of air bubbles. Membranes were prepared by a phase inversion method, namely by immersion precipitation, at room temperature. The procedure consists of casting the obtained homogeneous solution onto a glass support with a casting knife (gap 200 μm), followed by its direct immersion into a coagulation bath containing water. Membranes precipitate instantly due to solvent exchange with water (acting as a non-solvent), producing as a product a white, thin, flat porous film that was kept in water for 24 h to ensure complete solvent removal and then dried in air for 48 h before storage.

For the ILs incorporation, 3 × 3.5 cm^2^ membrane samples were placed in an oven at 60 °C for 72 h. Dry membranes were weighed using an analytical scale (Sartorius ED224S Extend Analytical Balance) and immersed into a 75% *w*/*w* solution of ionic liquid and milliQ^®^ water for 72 h. After that, the membranes were blot dried and placed in the oven at 60 °C for 72 h to ensure water removal and then reweighed (see Table 2). The difference in weight before and after soaking corresponds to the amount of ILs introduced to the membrane.

Fourier transform infrared spectroscopy (FT-IR) was used to verify the presence of the ILs in modified membranes and to analyze the membrane after CO_2_ sorption studies. Spectra were obtained using a Bruker Vertex-70 (Madrid, Spain) instrument with an attenuated total reflectance (ATR) sample holder by acquiring 32 cumulative scans with a resolution of 4 cm^−1^.

Physical characterization of the membranes was performed by environmental scanning electron microscopy (ESEM) using a FEI Quanta model 600 electron microscope (Brno, Czech Republic) with a resolution of 3 nm. Membranes were fractured in liquid nitrogen fixed to the suitable support for the cross-section analysis. Morphological characterization was carried out under a low vacuum with a large feed detector (LFD). The ESEM micrographs were recorded with 600 magnifications. Micrographs obtained from ESEM were analyzed with ImageJ software to determine membrane thickness and macrovoid size. Moreover, elemental mapping on the cross-sectional micrographs was obtained using PentaFETx3 Link Dispersive Energy *X*-ray Spectroscopy (EDXS) managed by Inca Oxford installed in the same equipment.

The supporting membranes porosity (N_PSU and D_PSU) (*ε*) was determined from the bulk and the PSU density by using the following Equation (1):(1)$$\varepsilon =\left(1-\frac{\rho_{m}}{\rho_{psf}}\right)*100\%$$
where *ρ_m_* and *ρ_psf_* correspond to the membrane and PSU density (1.24 g/cm^3^), respectively [18].

IL interaction with the membrane surface was determined by contact angle measurement with a Dataphysics OCA 15EC. MilliQ^®^ water and 75% *w*/*w* solutions of ionic liquid in milliQ^®^ water were placed in a Hamilton de 500/gt syringe. A 3 μL droplet of the studied liquid was placed on the top and bottom surface of each membrane type, as both surfaces were in contact with the soaking solution. The contact angle was calculated from a digital image using the SCA software included in the apparatus as the average of 3 measurements.

The membrane surface roughness was analyzed by atomic force microscopy (AFM) using an Agilent 5500 Scanning Probe Microscope in contact mode with a Multi 75 Al-G Budget Sensors tip (resonance frequency 75 kHz and force constant 3 N/m) combined with WsxM software [24]. The surface roughness parameters of the membranes expressed in terms of mean roughness (Rα) and root mean square of Z (Rԛ) were obtained by a roughness analysis calculated by the software.

### 2.4. CO_2_ Sorption Studies

The CO_2_ solubility in the obtained membranes was determined by its pressure decay in a closed chamber. All tests were carried out in a stainless-steel reactor of 50 mL volume with a glass filler to reduce the total volume of the system to 28 mL. All experiments were performed at 30 °C to ensure reproducibility. The pressure was acquired by an electronic manometer (STORKSolutions, model UPS-HSR-B02P5G, range −1 to 2.5 bar). The system used for the CO_2_ sorption studies is depicted in Figure 3 (real image in Appendix A). The procedure for conducting the experiments is as follows. The membrane (3 × 3 cm^2^), previously dried at 60 °C for 72 h and stored under non-humidity ambient conditions until ambient temperature, was introduced into the reactor. Then, the reactor was immersed in the water bath. After temperature stabilization, the system was purged with 2 barg of pure N_2_ three times to ensure complete air removal from the reactor volume. The next step was pressurizing the system to 2 bar by closing the valves with pure N_2_ for 30 min until pressure stabilization was reached to confirm its hermeticity. Then, the system was depressurized, purged, and charged with pure CO_2_ to 2 barg. The pressure decay was monitored for 5 h. Finally, the system was vented and the membrane extracted for further characterization with ATR-IR. Blank experiments were conducted using N_2_ instead of CO_2_.

The solubility coefficient is given in m^3^ of gas at standard conditions (1 atm and 273.15 K (STP)) per m^3^ of membrane conditions, obtained by the following Equation (2):(2)$$S=\frac{V_{stp}}{V_{m}*P_{f}}$$
where *S* corresponds to the solubility coefficient [m^3^ (STP) m^−3^_membrane_ atm^−1^], *V_stp_* is the volume of CO_2_ adsorbed in STP [m^3^], *V_m_* corresponds to membrane volume [m^3^], and *P_f_* to the final pressure of the system after stabilization [atm].

Further, the CO_2_ sorption was recalculated to moles of CO_2_ adsorbed per mol of IL to normalize and compare the results. The CO_2_ solubility in the unmodified PSU membrane (D_PSU 6.28 × 10^−5^ mol CO_2_/g PSU and N_PSU 2.8 × 10^−4^ mol CO_2_/g PSU) was subtracted in order to study the influence of supporting the ILs on its CO_2_ sorption capacity. The adsorbed CO_2_ moles were determined following Equation (3):(3)$$nCO_{2}=\frac{\left(P_{i}-P_{f}\right)\times \left(V_{v}-V_{m}\right)}{R\times T}$$
where *n*CO_2_ corresponds to adsorbed CO_2_ moles [mol], *P_i_* and *P_f_* correspond to the initial and final pressure [Pa], respectively, *V_v_* is the volume of the empty chamber (gas volume) [m^3^], *V_m_* is the volume of analyzed material [m^3^], *R* is the gas constant (8314 [m^3^⋅Pa⋅K^−1^⋅mol^−1^]), and *T* is the temperature [K]. The volume of the material was measured based on the correlation between the PSU volume, taking into account the membrane porosity and ILs volume in each membrane calculated from the mass increase, and the density of the ILs.

## 3. Results

### 3.1. Synthesized Ionic Liquids

Following the procedure listed in 2.2, a white crystalline solid was obtained as a product, which was then filtered and dried under vacuum leading to 480 g of 1-buthyl-3-methylimidazolium chloride (BMI.Cl) with a 96% yield. The product was characterized by ^1^H NMR (see Appendix A). After BMI.Cl obtention, the anion exchange was performed to acquire the desired task-specific ILs (Table 2). The anion exchange process provided the IL structures in quantitative yields (>99%). The products obtained displayed the expected signals in the ^1^H NMR spectra (see Appendix A) and IR spectra (see Appendix A).

The prepared ILs were divided into 3 groups to study the influence of different anions of the BMI.X on the CO_2_ capture rate: (1) ILs bearing carboxylate anions (BMI.FO, BMI.PIV, and BMI.BENZ), (2) ILs bearing dicarboxylate anion (BMI.MAL), and (3) ILs bearing carboxylate anions containing amine (BMI.PRO).

### 3.2. Physio-Chemical Characterizations of Membranes

To determine the efficiency of the incorporation of ILs in the membranes of different internal morphologies and to study their CO_2_ adsorption capabilities, two blank membranes were prepared (D_Blank and N_Blank) and used as a support for 5 different ILs. A total of 12 different membranes were prepared (Table 2).

The internal membrane morphology was determined by a cross-sectional study using environmental scanning electron microscopy (ESEM). Figure 4 and Figure 5 show the obtained micrographs. As can be seen, the internal structure of the D_Membranes (Figure 4) was filled with micropores and macrovoids with a drop-like structure, while the N_Membranes (Figure 5) were filled with macrovoids of finger-like structure. This morphology changes because of the solvent used for polymeric solution preparation due to different interaction of the solvents with water when the casted polymeric solution was immersed into the coagulation bath during membrane preparation. The polymer precipitation velocity is one of the key factors responsible for the final membrane morphology. The faster the penetration of the coagulation agent, the higher the asymmetry obtained, including more elongated macrovoids and a thicker membrane. Several parameters influence the precipitation velocity, such as precipitation kinetics, polymeric solution viscosity, and the thickness of the casting knife.

Comparing the images of blank membranes with the ones obtained from the membranes containing ILs, we confirmed that no structural changes in the membrane were induced by incorporating ILs into its structure. The authors suspect that the ILs were adsorbed only by physical interactions.

Table 3 shows the membrane preparation conditions and the obtained membrane characteristics. In our case, a higher thickness value was obtained with N_Membranes. Such membranes show an internal morphology mainly composed of an asymmetric fingerlike structure on the top portion due to a faster polymer precipitation. The N_Membranes exhibit a higher porosity and thickness, but a lower macrovoid size than the D_Membranes.

The membrane thickness has also been examined in the membranes with ILs to determine if the soaking process promotes an expansion of the membrane, increasing the thickness. Figure 6 shows that the membrane thicknesses for the N_Membranes have similar values (between 130 μm and 115 μm), and the same applies to the D_Membranes (between 90 μm and 100 μm).

#### 3.2.1. Infrared Studies (IR)

IR studies were performed to confirm the presence of ionic liquids in the membranes. Figure 7 and Figure 8 show a part of the stacked IR spectra in which the unmodified membrane (D_PSU and N_PSU), the corresponding ionic liquid (BMI.BENZ), and the modified membrane (D_BMI.BENZ and N_BMI.BENZ) can be seen. The most characteristic bands that can confirm the presence of ionic liquids in the membrane were the ones that correspond to the stretching of C-N (from the cation) that can be seen at around 1640 cm^−1^, and the presence of bands corresponding to the symmetrical and asymmetrical stretching carbonyl simple bond (from the carboxylate anions) that can be seen at around 1360 cm^−1^ and 1570 cm^−1^, respectively. No new chemical bonds between the ionic liquid and the PSU have been detected, confirming that the ionic liquids were physically incorporated into the membrane by pore and macrovoid filling and not by chemical reaction. Even though the ATR-IR only penetrates the first 4 μm of the membrane, we were able to confirm the presence of ILs in the membrane structure. (For all spectra, see Appendix A).

#### 3.2.2. Energy Dispersive *X*-ray Elemental Analysis

The energy dispersive *X*-ray elemental analysis was performed on the cross-section of the membranes. The presence of nitrogen inside the membranes was assigned to the imidazolium cation and confirmed the presence of an ionic liquid. Figure 9 shows an example of elemental analysis characterization, the D_BMI.BENZ membrane cross-section, which highlighted a good dispersion of nitrogen (blue) all along the membrane and confirmed the presence of ionic liquids inside the membrane. It must be noted that the ionic liquid was well dispersed and not concentrated in the macrovoids (for all micrographs, see Appendix A and Appendix A). Besides the nitrogen, carbon, sulfur, and oxygen was found to be present, mainly in PSU structure.

### 3.3. Sorption of the ILs into Membrane Pores by Capillary Forces

#### 3.3.1. Mass Increase Determination

The amount of ILs introduced into the membrane was determined by the mass change before and after the soaking process. Figure 10 and Figure 11 display the results of the mass increase (%) obtained for the membranes of different structures.

As can be seen in Figure 10 and Figure 11, the best adsorption of IL in the membrane was achieved with BMI.PIV and BMI.BENZ in both types of membranes. In these two cases, the mass increase rose to values near 200% for the D_Membranes and 250% for the N_Membranes, while for the other ILs, the increase was lower than 50% for the D_Membranes and lower than 90% for the N_Membranes, indicating that these ionic liquids can be efficiently adsorbed. This behavior has been ascribed to the higher affinity of the relatively hydrophobic polysulfone polymer by the ILs with more hydrophobic character, such as BMI.PIV, and those ILs with aromatic anions, such as BMI.BENZ, could plausibly cause π-π stacking interactions with polysulfone aromatic rings.

As expected, the best adsorption results were obtained with the N_Membranes for all the studied ILs. The introduction of the ILs to the membrane pores by soaking is based on capillary action. Capillary action is explained by Jurin’s law, which states that the maximum high (*h*) of a liquid in a capillary tube is inversely proportional to the tube diameter. If the diameter of the tube (in our case, of the macrovoid) is sufficiently small, then the combination of surface tension and adhesive forces between the liquid and the pore wall act to propel the liquid. The law is defined by the following equation:(4)$$h=\frac{2ϒcos\theta }{\rho gr}$$
where *h* corresponds to the height of a liquid in a column (macrovoids), ϒ to the surface tension of each ILs (mN/m), *ρ* to the density of the ionic liquid (g/mL), *g* to the local acceleration due to gravity, *r* to the tube or macrovoids radius, and *θ* is the contact angle between the membrane and the ILs. Based on the capillary action theory, the N_Membranes should exhibit a higher efficiency of ILs sorption, as it has a higher contact angle (*cos θ*, Figure 15) and a lower macrovoid size (*r*, Table 3). Moreover, the N_Membranes have a higher porosity and thickness (Table 3), which allows them to retain higher volumes of ILs.

As the incorporation of the ILs was achieved mainly by capillary forces, it has been attempted to correlate the difference in soaking efficiency among the ionic liquids used with the viscosity of the ionic liquids, or rather the viscosity of the solutions used for soaking (75%wt), their density, and their surface tension. It was expected that a higher solution surface tension (ϒ) or lower density (*ρ*) would result in a higher quantity of ILs introduced to the membrane. However, no direct correlation has been discovered. All obtained values can be found in SP (specifically Appendix A).

#### 3.3.2. Membrane Roughness Determination

Membrane roughness also plays a role in the absorption efficiency of the ionic liquids. Figure 12 and Figure 13 show a 3D representation of the D_PSU and N_PSU membrane surfaces, respectively. In the images, it can be seen that the top surfaces were more homogeneous those on the bottom, in both cases.

Table 4 includes the roughness values obtained for both types of supporting membranes used in this work. D_PSU has similar Ra and Rq values for both surfaces, while the surfaces of N_ PSU were diverse, as the top surface was 5 times rougher than the bottom surface. Moreover, comparing the roughness of each surface between the membranes, it can be seen that N_ PSU has higher values for the top surfaces and D_ PSU for the bottom surfaces.

In addition, the skewness of the membranes was studied to provide more information about the membrane surfaces. Positive values correlate with surfaces where the roughness was constituted by hills, while negative values correspond to a surface with more valleys. The membrane prepared with NMP (N_ PSU) was the only one among all the membranes studied that presented a positive skewness value on its top surface. The positive skewness value can be related to facilitated capillary action and may contribute to the higher mass increase during the membrane soaking.

#### 3.3.3. Contact Angle Determination

In order to determine the influence of the surface–liquid interaction on the ionic liquid absorption behavior, contact angle experiments were carried out on the bottom and top surfaces of the supporting membranes and soaking solutions. Figure 14, Figure 15 and Figure 16 show the results obtained for the CA measurements, including milliQ^®^ water as a reference and the membrane mass increase after soaking (for contact angle images, see Appendix A). All values of CA are below 90°, which indicates that the surfaces were IL-philic. Thus, they will adsorb ILs or will let them pass. We demonstrated that the lower the contact angle between the soaking solution and the membrane, the more efficient the soaking procedure. As can be seen in Figure 14 and Figure 15, corresponding to the top surfaces of both membranes, the solutions that have the lowest CA were the ones prepared with BMI.PIV and BMI.BENZ; thus, they ones that resulted in the highest mass increase. Nevertheless, no differences in the CA values were observed between the bottom surfaces (Figure 16 and Figure 17), suggesting that the ILs absorption could be mainly influenced by the properties of the top surface of the membranes.

As for the influence of the membrane type, in general, lower contact angles were obtained with the membranes prepared with DMF as a solvent, but a higher mass increase was obtained with the membranes prepared with NMP. However, as previously mentioned, due to lower macrovoid size, stronger capillary forces act on the N_Membranes, and therefore, the membrane was more predisposed to adsorb higher amounts of ILs.

Based on the soaking studies and the characterization of both types of membranes, we can conclude that in general better membranes for IL absorption using the soaking method were the N_Membranes due to their high porosity and top surface properties.

### 3.4. Solubility Results

The sorption of N_2_ and CO_2_ of all the studied samples was evaluated; however, no sorption was detected in the case of N_2_. These results suggest a >99% selectivity for CO_2_ sorption. Table 5 shows the results obtained for CO_2_. Nevertheless, an increase in CO_2_ sorption capabilities by at least one order of magnitude was achieved for all modified membranes in comparison to the pristine PSU membranes (Table 5). However, the results obtained for each ILs are not comparable, as different amounts were introduced to the membrane support.

For another point of view, CO_2_ solubility was determined to assess the direct effect of supporting the ILs in a PSU matrix. These results were obtained by normalizing the absorbed CO_2_ moles in the supported ionic liquid by the subtraction of CO_2_ sorption in the blank membranes to obtain the mol of CO_2_ sorbed by the mol of the IL (see Section 2.4). In general, the CO_2_ sorption results (Figure 18) revealed an enhancement of the IL sorption capacity by the use of polysulfone as a polymeric matrix for its support. This behavior could be ascribed to an expansion of the contact between the ILs and the gas. The numerical values obtained for the CO_2_ moles adsorbed and their solubility can be found in Appendix A.

As previously mentioned, the ILs used in this work were divided into three groups, and different improvements were established for each type. The first one (Type 1) contains a carboxylate anion and exhibits different behaviors for each compound. The best solubility among this group was obtained with BMI.PIV, supported in both membrane types, doubling the CO_2_ solubility. Taking into account that this IL was introduced in high amounts to the membranes, it also shows the highest capacity of total CO_2_ adsorption per surface of the membrane. While BMI.FO was supported in the N_Membranes and BMI.BENZ in the D_Membranes, these membranes were the only ones that exhibited roughly the same CO_2_ sorption as the bulk ionic liquid.

The second type of ionic liquid, the one containing the dicarboxylate anion (Type 2), yielded the highest adsorption value for both supported and unsupported ILs among all studied systems. The values obtained for supported BMI.MAL were twice as high as those for the unsupported one, reaching 0.24 mol_CO2_/mol_IL_.

The third type of ILs was the one containing an amine group in the anion BMI.PRO (Type 3). In this case, the CO_2_ solubility studies showed the lowest efficiency in the bulk ILs. However, when supported, the value increased 10 times. This IL has the highest viscosity among those tested; thus, by supporting it, we were able to increase the contact surface area and thereby, increase the diffusion, as it is directly connected to the viscosity by the Stokes–Einstein law. This suggests that the use of a porous membrane to support ILs in order to improve their CO_2_ capture rate will have the highest impact in highly viscous ILs.

The ILs supported on membranes made with NMP (N_Membranes) yielded better results in most cases, due to the higher porosity of the membrane compared to that of the D_Membranes, as well as the same or better ILs distribution. A positive synergistic effect between the IL and the PSU matrix was found that improved the CO_2_ sorption capacities of both ILs and the membranes.

In order to determine the type of sorption present in our CO_2_ capturing system, ATR-IR analysis were carried out after the solubility studies. Since the obtained spectra did not show the creation of a carbamate bond, a physical CO_2_ sorption must be expected. However, chemical CO_2_ sorption in ILs cannot be discounted due to the low amount of the ILs compared to the PSU matrix, which could quench the new IR bands formed by the low amount of CO_2_ chemically incorporated into the membranes.

## 4. Conclusions

N_Membranes can uptake more ILs than D_Membranes, due to a higher free volume inside the membrane resulting from its porosity and thickness. Additionally, its characteristics, such as positive skewness or smaller macrovoids size, affect the strength of capillary action and provide an uptake advantage.

The incorporation of ILs was demonstrated by the mass change after soaking experiment and by the ATR-IR and ESEM-EDX analysis. Moreover, the incorporation was found to be physical, which is supported by ATR-IR studies, and it did not affect the membranes’ internal morphologies nor thicknesses.

The amount of the ILs introduced into the membrane depends on the membrane porosity and the ILs interaction with its surface, especially the contact angle. The ionic liquids BMI.PIV and BMI.BENZ were found to have the lowest contact angle on the polysulfone membranes, which allowed for easier penetration.

The CO_2_ solubility was improved when ILs were supported into porous membranes thanks to improvement in the contact area between the liquid and gas. This phenomenon was specially noted with ILs of high viscosity where dispersion of ILs in the pores caused a more pronounced improvement in diffusion. The best results were obtained with BMI.MAL, which possesses a double carboxylate anion. Finally, we can confirm that the N_Membranes were the most suitable in terms of ILs uptake and CO_2_ solubility improvement. A positive synergistic effect between the IL and the PSU matrix was found, which improved the CO_2_ sorption capacities of both the ILs and the membranes.

## Figures and Tables

**Figure 1 polymers-14-04865-f001:**
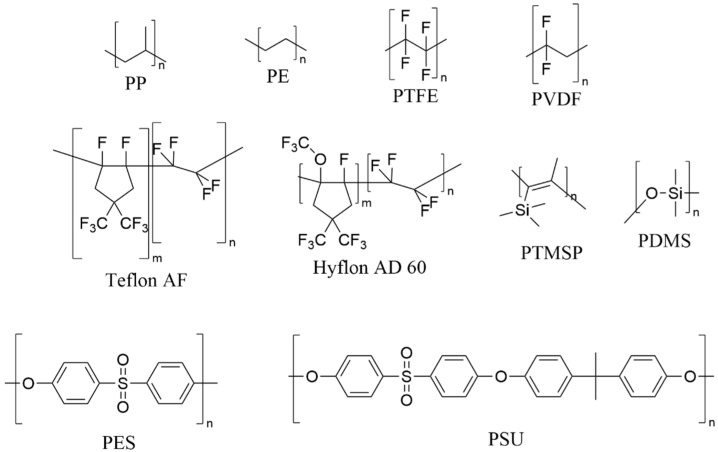
Polymer for polymeric membrane preparation; PP: polypropylene; PE: polyethylene; PTFE: polytetrafluoroethylene; PVDF: polyvinylidene fluoride; Teflon amorphous fluoropolymers; Hyflon amorphous perfluoro polymer; PTMSP: poly (1-trimethylsilyl-1-propyne); PDMS: polydimethylsiloxane; PES: polyethersulfone; PSU: polysulfone.

**Figure 2 polymers-14-04865-f002:**
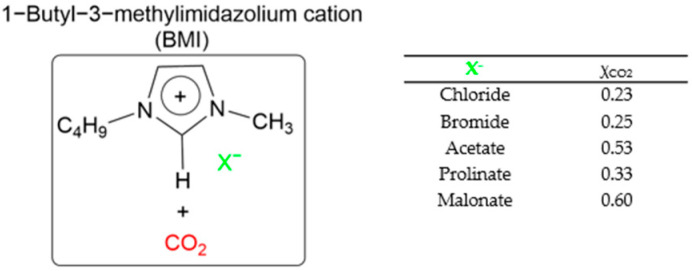
CO_2_ solubility values reported by the molar fraction of CO_2_ at 25 °C and 10 bar [11].

**Figure 3 polymers-14-04865-f003:**
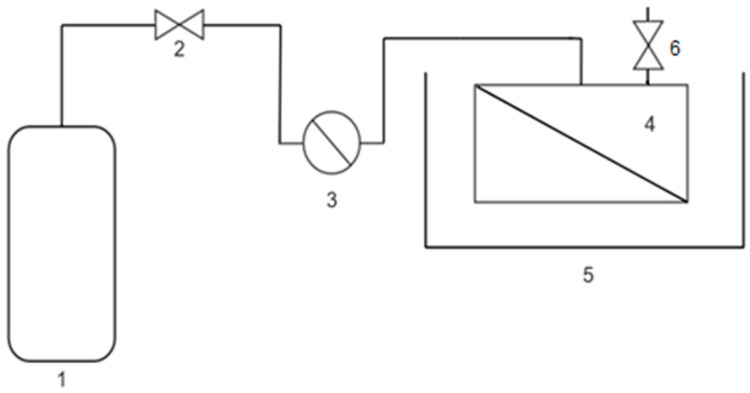
Solubility system: (1) CO_2_ bottle; (2) valve to close the system; (3) electronic manometer; (4) reactor containing membrane; (5) water bath at 30 °C; (6) exit valve.

**Figure 4 polymers-14-04865-f004:**
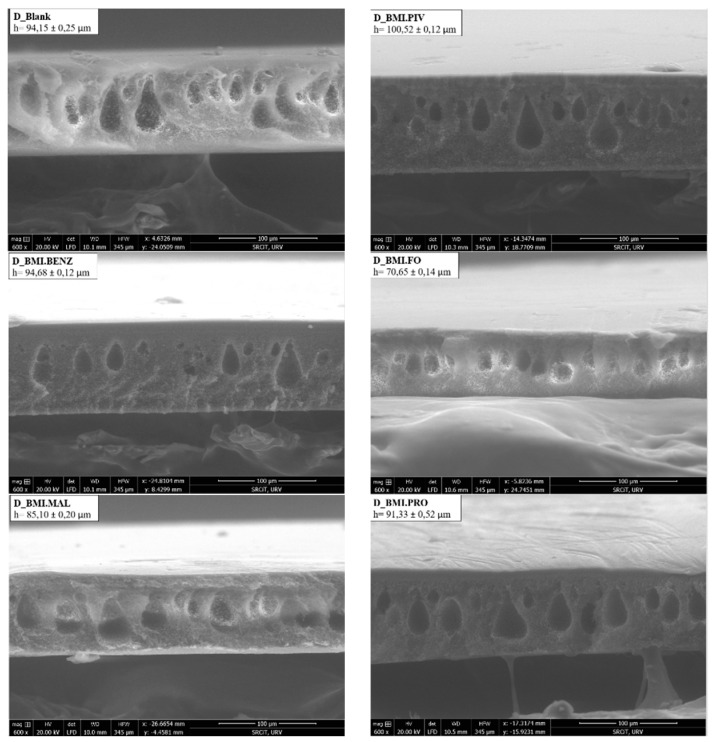
ESEM micrographs of the DMF membranes.

**Figure 5 polymers-14-04865-f005:**
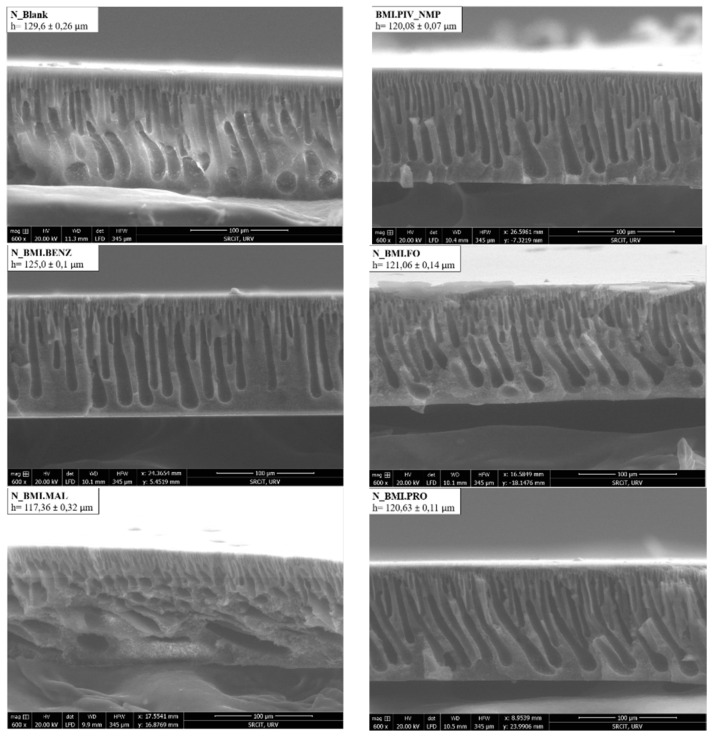
ESEM micrographs of the NMP membranes.

**Figure 6 polymers-14-04865-f006:**
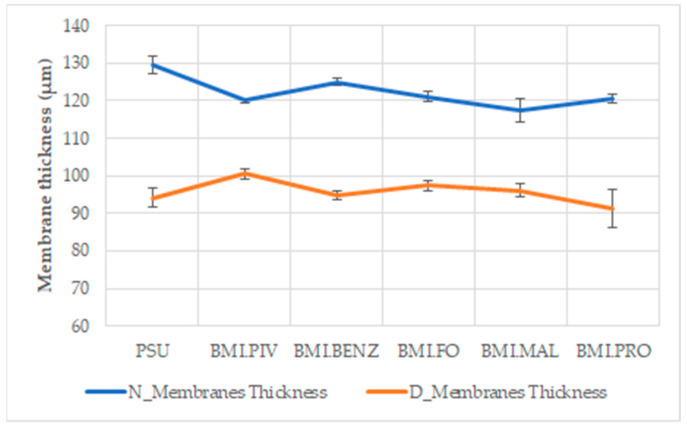
Membrane thickness after soaking, as measured by ESEM analysis.

**Figure 7 polymers-14-04865-f007:**
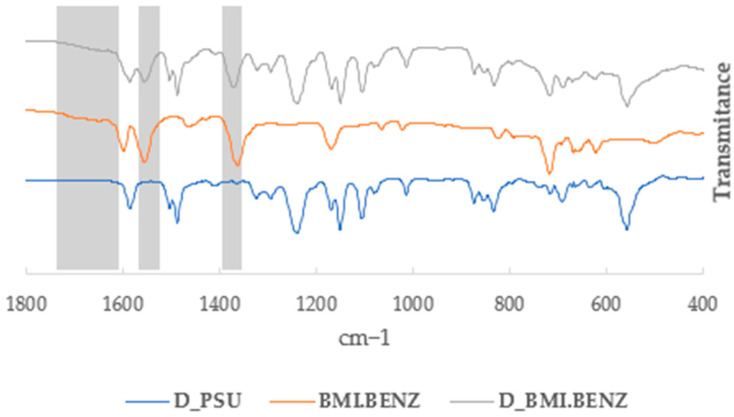
Partial IR spectra of D_PSU, BMI.BENZ IL, and D_BMI.BENZ.

**Figure 8 polymers-14-04865-f008:**
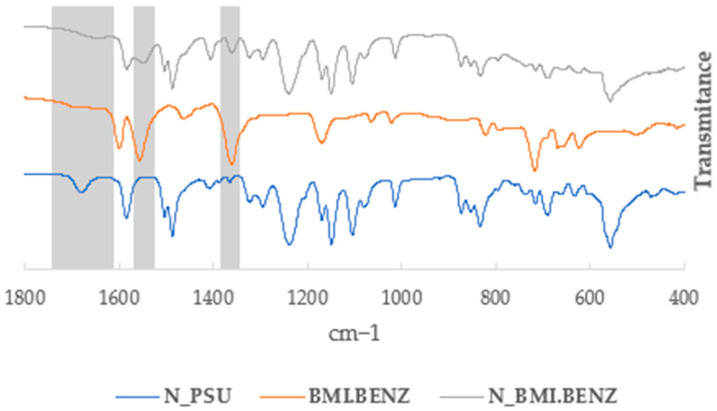
Partial IR spectra of N_PSU, BMI.BENZ IL, and N_BMI.BENZ.

**Figure 9 polymers-14-04865-f009:**
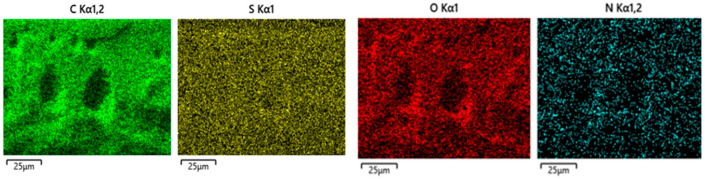
EDX analysis of D_BMI.BENZ.

**Figure 10 polymers-14-04865-f010:**
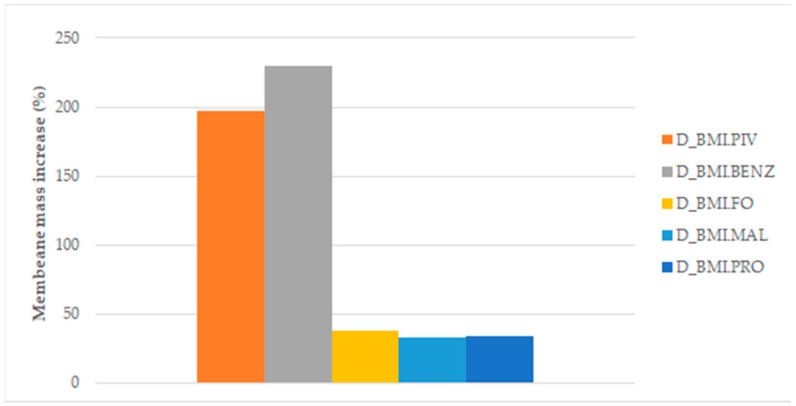
Mass increase after soaking the DMF membranes.

**Figure 11 polymers-14-04865-f011:**
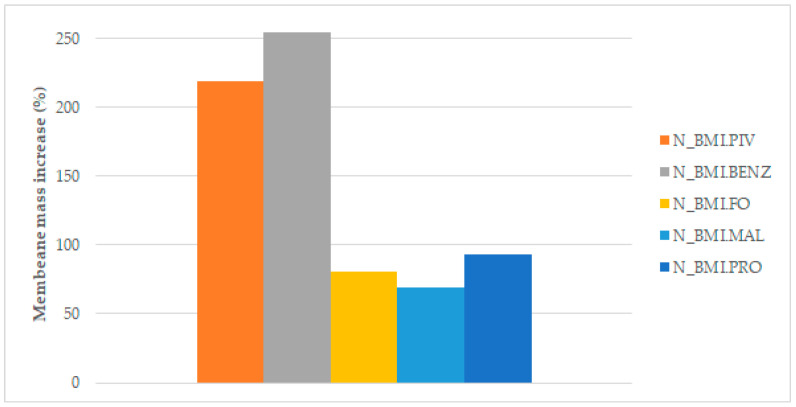
Mass increase after soaking the NMP membranes.

**Figure 12 polymers-14-04865-f012:**
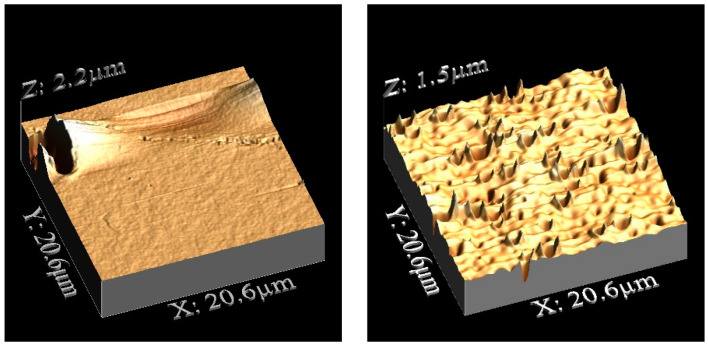
(**Left**) AFM top surface, D_ PSU. (**Right**) AFM bottom surface, D_ PSU.

**Figure 13 polymers-14-04865-f013:**
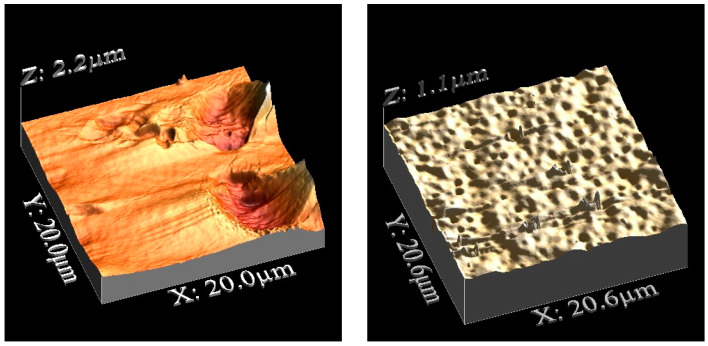
(**Left**) AFM top surface, N_ PSU. (**Right**) AFM bottom surface, N_ PSU.

**Figure 14 polymers-14-04865-f014:**
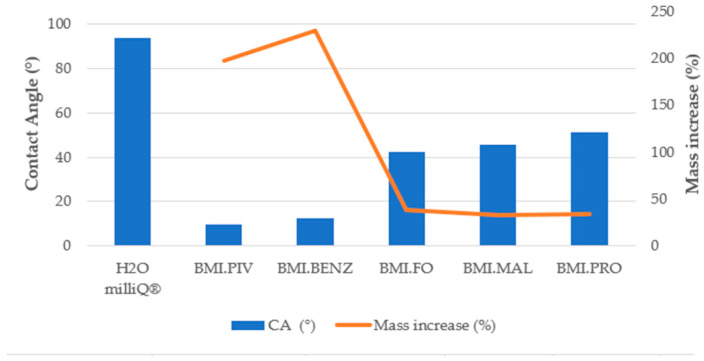
Top surface contact angle and membrane mass increase correlation for the D_Membranes.

**Figure 15 polymers-14-04865-f015:**
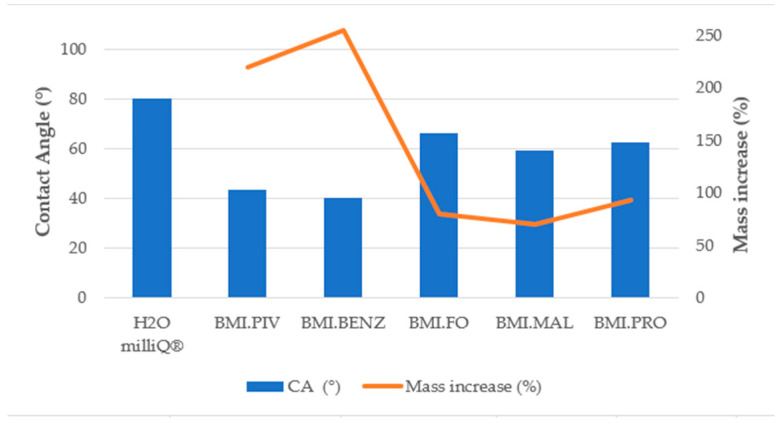
Top surface contact angle and membrane mass increase correlation for the N_Membranes.

**Figure 16 polymers-14-04865-f016:**
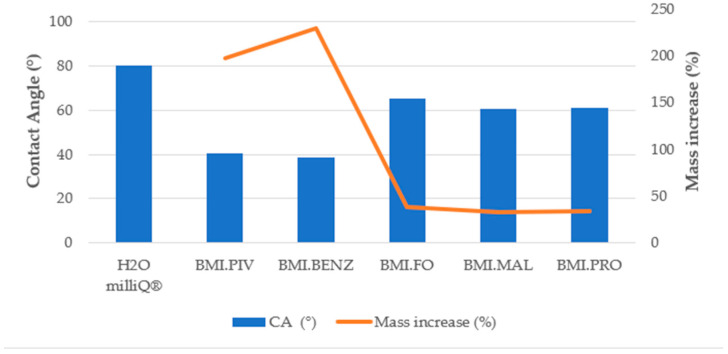
Bottom surface contact angle and membrane mass increase correlation for the D_Membranes.

**Figure 17 polymers-14-04865-f017:**
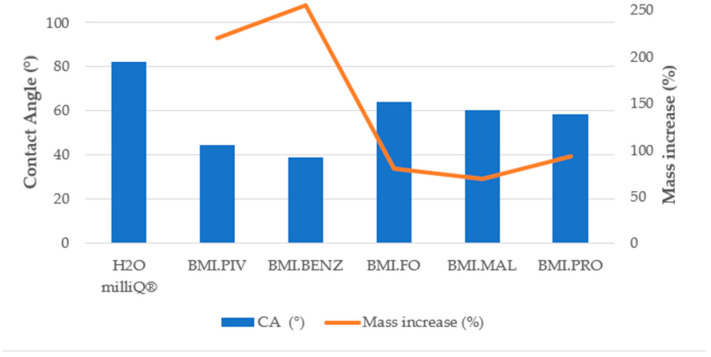
Bottom surface contact angle and membrane mass increase correlation for the N_Membranes.

**Figure 18 polymers-14-04865-f018:**
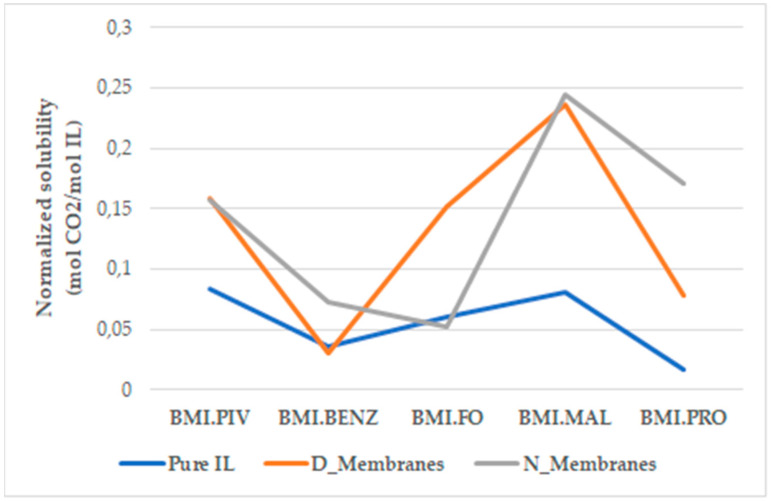
Solubility results for bulk ionic liquids and supported ionic liquids.

**Table 1 polymers-14-04865-t001:** Synthetized ionic liquids.

Type	Ionic Liquid	Carboxylic Acid Used	Ionic Liquid Structure
1	BMI.PIV	Pivalic Acid (PIV)	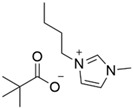
1	BMI.BENZ	Benzoic Acid (BENZ)	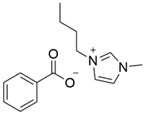
1	BMI.FO	Formic Acid (FO)	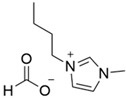
2	BMI.MAL	Malonic Acid (MAL)	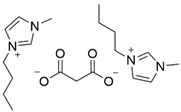
3	BMI.PRO	Proline (PRO)	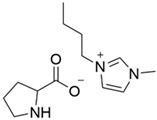

**Table 2 polymers-14-04865-t002:** Supported ionic liquid membranes.

Classification	Sample	Solvent	Ionic Liquid Cation	Ionic Liquid Anion
D_Membranes	D_PSU	DMF	-	-
	D_BMI.PIV		BMI	Pivalate
	D_BMI.BENZ		BMI	Benzoate
	D_BMI.FO		BMI	Formate
	D_BMI.MAL		2 BMI	Malonate
	D_BMI.PRO		BMI	Prolinate
N_Membranes	N_PSU	NMP	-	-
	N_BMI.PIV		BMI	Pivalate
	N_BMI.BENZ		BMI	Benzoate
	N_BMI.FO		BMI	Formate
	N_BMI.MAL		2 BMI	Malonate
	N_BMI.PRO		BMI	Prolinate

**Table 3 polymers-14-04865-t003:** Membrane preparation conditions and characteristics.

	D_Blank	N_Blank
Process	Immersion precipitation	
Polymeric Solution	20% *w*/*w* of polysulfone	
Solvent	DMF	NMP
Casting Knife (μm)	200	
Support	Glass	
Membrane Thickness (μm)	94.1 ± 0.2	129.6 ± 0.3
Porosity (%)	67	73
Macrovoid size (μm)	32 ± 7	15 ± 4

**Table 4 polymers-14-04865-t004:** Atomic force microscopy results for D_ PSU and N_ PSU membranes.

Membrane	Surface	Average Roughness (Ra)	Root Mean Square (Rq)	Skewness
D_ PSU	Top	0.05 ± 0.6	0.13 ± 0.08	−0.84
D_ PSU	Bottom	0.0654 ± 0.007	0.106 ± 0.004	−2.29
N_ PSU	Top	0.16 ± 0.04	0.21 ± 0.02	0.27
N_ PSU	Bottom	0.027 ± 0.005	0.041 ± 0.002	−3.69

**Table 5 polymers-14-04865-t005:** Solubility coefficients obtained by Equation (2).

Membrane	Solubility Coefficient (*S*) (m^3^(STP) m^−3^_membrane_ atm^−1^)
D_PSU	4.16 × 10^4^
D_BMI.PIV	5.25 × 10^6^
D_BMI.BENZ	2.72 × 10^6^
D_BMI.FO	2.03 × 10^5^
D_BMI.MAL	1.31 × 10^5^
D_BMI.PRO	7.27 × 10^4^
N_PSU	1.83 × 10^5^
N_BMI.PIV	7.30 × 10^11^
N_BMI.BENZ	3.40 × 10^11^
N_BMI.FO	5.08 × 10^5^
N_BMI.MAL	1.05 × 10^6^
N_BMI.PRO	2.52 × 10^6^

## Data Availability

The data presented in this study are available on request from the corresponding author.

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
