# Peer review of "Supported Imidazolium-Based Ionic Liquids on a Polysulfone Matrix for Enhanced CO2 Capture"

_polymers, 2022, doi:10.3390/polym14224865_

Round 1
Reviewer 1 Report
The publication entitled: „Supported imidazolium-based ionic liquids on polysulfone matrix for enhanced CO2 capture” presented for review is very interesting.
The work is likely to be of interest to a large group of scientists dealing with ionic liquids and environmental problems.
The methodology of the planned research does not raise any objections, and the obtained results have been correctly interpreted.
I recommend this work to be accepted for publication in Polymers.
I only have minor comments: the anion cannot be dicarboxylic/ carboxylic but dicarboxylate/ carboxylate, in Figure 1, the R substituents in the formulas should have superscript rather than subscript numbers, because the subscript number tells about the number of groups.
The authors write that ionic liquids are a combination of an organic cation with an organic or inorganic anion, which is not true. In the literature [Shobukawa H., Tokuda H., Tabaata S., Watanabe M., Electrochim. Acta, 50, 1-5 (2004).] ionic liquids with inorganic anions are described. Completely inorganic ionic liquids are also known [Dai L., Yu S., Shan Y., He M., Eur. J. Inorg. Chem. 237-241 (2004)]. The correct definition of ionic liquids is given by P. Wasserscheid who is an authority in the field of ionic liquids: „Ionic liquids are salts that are liquid at low temperature (< 100oC) which represent a new class of solvents with nonmolecular, ionic character”. [Wasserscheid P., Keim W., Angew. Chem. Int. Ed., 39, 3772-3789 (2000)].
Author Response
Thank you very much for the positive revision. As suggested, the names of the anions were change from be dicarboxylic/carboxylic to dicarboxylate/carboxylate along the text. Due to the issues with copyright permissions Figure 1 will be removed from the manuscript. Instead, we will place a new figure centered only in BMI cation with a table containing values of CO2 sorption of ILs reported in literature. All recommended articles concerning the nature of ions and definition of ionic liquids were incorporated and clarifications were done, see lines 65-75. We hope that the current version of the revised document will find your approval.
Reviewer 2 Report
This work deals with the CO2 capture using polysulfone polymeric matrix supported ionic liquids. The influence of the membrane structure and the nature of ILs on the CO2 capture abilities were investigated. I believe the data is exciting and it could be useful for people closer to Membrane Science. The manuscript is very well organized, and a complete synthesis procedure has been described in detail. The characterization of the synthesized material has been done using NMR and IR that is sufficient. Authors mainly focusses on 1H NMR. Is there any reasons authors did not perform C NMR? If possible, please provide C NMR results as well. How authors determined the purity of the synthesized products like ionic liquids? It is not clear from the manuscript. I suggest authors to determine the purity. The main evaluation results are also comprehensively studied using various techniques such as contact angle, Energy dispersive X-Ray elemental analysis, membrane roughness through AFM, and solubility. I could not find any contradictory results as all results are supporting each other. The discussion section is also well explained and I have no major comments on that. My only concern is on the synthetic part and the manuscript can be accepted after addressing those comments.
Author Response
Thank you very much for the positive revision. Answering the doubts concerning purity control of ILs and use of 1H NMR only: The purity of the ILs was determined by 1HNMR, which is a common method when studying ILs or other organic compounds. The chemical structures were identified and the possible impurities, such us unreacted materials, were not detected. This statement was clarified in line 167-169. 13C NMR is very useful tool when studying synthesis of new materials especially in combination with 1H NMR analysis. However, ILs used in the study are already reported in the literature, thus we assume that 1H NMR will be sufficient, as 13C NMR is more expensive and more time consuming. We hope that the current version of the revised document will find your approval.